# Older Age as a Predictor of Ongoing Active Changes in Follow-Up Cardiac Magnetic Resonance in Children with Acute Myocarditis

**DOI:** 10.3390/jcm13216498

**Published:** 2024-10-30

**Authors:** Łukasz A. Małek, Anna Gwiazda, Marzena Barczuk-Falęcka

**Affiliations:** 1Department of Nursing, Faculty of Rehabilitation, University of Physical Education, 00-968 Warsaw, Poland; 2Department of Pediatric Radiology, Pediatric Hospital of the Medical University of Warsaw, 02-091 Warsaw, Poland

**Keywords:** edema, mapping, late gadolinium enhancement, risk factors, prognosis

## Abstract

**Background/Objectives**: Cardiac magnetic resonance (CMR) is used to diagnose and monitor the course of acute myocarditis in adults and children. This study aimed to assess the frequency of persistent inflammation at follow-up CMR and to look for predictors of ongoing active changes in CMR in children with myocarditis. **Methods**: This retrospective study included 31 children (median age 15 years, 68% male) with clinically and CMR-diagnosed acute myocarditis who underwent baseline and follow-up CMR at a median of 6 months. Old and new Lake Louise criteria were compared. **Results**: A complete resolution of changes was observed in four patients (13%) at follow-up, according to both criteria. Seven patients (23%) presented ongoing active changes, and twenty (64%) showed a persistent scar according to the old Lake Louise criteria. When the new Lake Louise criteria were used, an additional two patients (6%) were found to have persistent active changes instead of a persistent scar. Patients with persistent inflammation (nine patients, 29%) were older than those who showed recovery. None of the patients below 14 years of age presented active changes on their follow-up CMR and all the patients who showed inflammation were between 14 and 17 years old. **Conclusions:** Pediatric myocarditis can lead to persistent active changes in CMR beyond a 6-month follow-up in over a fifth of patients. The application of new Lake Louise criteria further increases that number compared to the old criteria. The only predictor of persistent inflammatory changes in CMR is older age.

## 1. Introduction

Acute myocarditis is one of the most frequent diseases in pediatric cardiology, with two peaks of occurrence—the first one in neonates and infants and the second one in adolescents [1]. Myocarditis may complicate common upper respiratory or gastrointestinal infections [1]. Although the clinical course is usually benign, with recovery in the time of few weeks without a sequel, on occasion, the disease may lead to sudden cardiac death or acute and chronic heart failure [1]. Typical symptoms include fever and chest pain or signs of heart failure, which are accompanied by changes in resting ECG, including ST elevation or T-wave inversion in the corresponding leads [1,2,3,4]. Cardiac necrotic markers, as well as markers of inflammation and brain-natriuretic peptide levels, are usually markedly elevated.

The final diagnosis is currently based on the results of cardiac magnetic resonance (CMR), which shows typical changes including diffused or regional edema on T2 sequences (T2 mapping and/or T2-weighted images) and signs of cardiomyocyte injury on T1 sequences (T1 mapping and/or late gadolinium enhancement) [1,2,5,6]. CMR has been shown to accurately depict acute myocarditis in comparison to the gold standard of endomyocardial biopsy without a need for the invasive test in most cases [6]. CMR is also used in follow-up to confirm the cessation of active changes, especially before a full return to physical activity.

We and others have demonstrated that persistent inflammation in CMR can be detected in over one-fourth of patients at the usual follow-up timeframe of around 6 months, despite the complete cessation of symptoms and the normalization of cardiac markers, an electrocardiogram (ECG) and echocardiographic pictures [7,8,9]. The clinical significance of this phenomenon is unknown and remains a subject of investigation [1]. However, current practice is to withhold children presenting such prolonged healing from sports activity until a second follow-up later on, which usually shows a cessation of inflammation.

Most of the studies so far have used old Lake Louise criteria to detect myocarditis based on T2-weighted images and LGE [9,10,11,12]. Our study aimed to shed more light on the persistence of inflammatory changes in CMR in children with acute myocarditis with the use of the more accurate new Lake Louise criteria based on parametric imaging (T1 and T2 mapping). The additional aim was to look for predictors of ongoing active changes in CMR, which can help to schedule the best time for a CMR follow-up to avoid unnecessary repeated examinations and gadolinium contrast administration in children.

## 2. Materials and Methods

### 2.1. Study Group

This retrospective analysis included 31 children [Caucasian, median age 15 years (interquartile range, IQR, 13.3–16.8 years), 68% male] hospitalized in the Pediatric Hospital of the Medical University of Warsaw between January 2021 and November 2023 with suspected myocarditis who had a CMR at baseline and underwent CMR follow-up at around 6 months from their initial scan. A COVID-19 infection or the administration of a COVID-19 vaccine as a reason for myocarditis were excluded. The diagnosis of acute myocarditis was based on the clinical picture and the result of a baseline CMR.

### 2.2. MR Protocol and Analysis

This study was performed with the use of a Siemens Sola 1.5 Tesla scanner (Siemens, Erlangen, Germany). The protocol included initial scout images, followed by cine-balanced steady-state free precession (bSSFP) breath-hold sequences in 2-, 3- and 4-chamber views. The short axis was identified using the 2- and 4-chamber images and a stack of acquired images which included the ventricles from the mitral and tricuspid valvular plane to the apex.

Pre-contrast T1 mapping with a modified Look-Locker sequence (MOLLI) and T2 mapping with a T2-prepared SSFP sequence were performed immediately after the acquisition of the bSSFP cine images and processed using MyoMaps software (Siemens, Erlangen, Germany). For that purpose, 3 short-axis slices (one basal, one mid-ventricular and one apical) and 2-, 3- and 4-chamber views were obtained. Subsequently, the acquisition of dark-blood T2-weighted (T2W) turbo spin echo images with fat suppression in the same orientations was achieved.

Following these acquisitions, 0.1 mmol/kg of a gadolinium contrast agent (gadobutrol—Gadovist^®^, Bayer Schering Pharma AG, Berlin, Germany) was administered and flushed with 15 mL of isotonic saline. Late gadolinium enhancement (LGE) images from 3 long axes and a stack of short-axis imaging planes were obtained with a breath-hold phase-sensitive inversion recovery (PSIR) sequence 5–15 min after the contrast injection. The inversion time was adjusted to null normal myocardium (typically between 250 and 350 ms, as assessed using a TI-scout acquisition).

Images were analyzed with the use of dedicated software (Syngo.via VB40, Siemens, Erlangen, Germany). All studies were assessed independently by two physicians—one cardiologist and one radiologist with expertise in cardiac MR. End-diastolic and end-systolic endocardial and epicardial contours were drawn semi-automatically for the left ventricle (LV) in the short-axis stack of bSSFP cine acquisitions. Delineated contours were used for the quantification of end-diastolic (LVEDVI) and end-systolic volume (LVESVI) indexed to the body surface area and ejection fraction (LVEF).

Pre-contrast T1 and T2 maps, as well as T2W images, were initially assessed visually for the presence of hyperintense areas. Pre-contrast T1 and T2 relaxation times and T2W signal intensity were calculated from 0.7 cm^2^ regions of interest (ROI) placed in all 17 segments. The segment consistently showing maximal values was used for the diagnosis and further calculations. The number of segments with an elevated T2W signal was also calculated.

The presence and type of LGE were assessed visually and divided into subepicardial, mid-wall or mixed. The location of LGE was described with division into the most common infero-lateral segments and other segments. The number of segments showing LGE was also considered. Abnormal native T1 and T2 values were defined as those greater than 1086 ms and greater than 52 ms, respectively, based on previously derived sequence- and scanner-specific cutoffs of 2 SDs above the respective means in a healthy pediatric population [13,14]. An increase in the myocardial T2 signal intensity ratio was defined as a signal intensity ratio of the left ventricular myocardium to skeletal muscle ≥2.0 [5]. Acute myocarditis was defined according to the original and new (known also as modified or updated) Lake Louise criteria [5,6].

### 2.3. Statistical Analysis

All results for categorical variables were presented as a number and a percentage. Continuous variables were expressed as a mean and standard deviation (SD) or median and interquartile range (IQR) based on the presence or lack of a normality of distribution, assessed by means of the chi-square test. To analyze the changes in parameters between the baseline and follow-up CMR in the whole studied group, a paired samples *t*-test or Wilcoxon test was used where appropriate. To search for predictors of ongoing active changes in CMR, the Mann–Whitney test was applied. All tests were two-sided with a significance level of *p* < 0.05. Statistical analyses were performed with MedCalc statistical software 10.0.2.0 (Ostend, Belgium).

## 3. Results

Details of baseline characteristics including symptoms, ECG changes, medications, and the timing of the follow-up CMR are presented in Table 1. The main symptom leading to admission was chest pain (71%), followed by fever (35%) and signs of heart failure such as dyspnea and fatigue (10%). ECG changes such as ST elevation or T-wave inversion were visible in 45% of patients. Baseline troponin I was markedly elevated in 30 patients (97%). All patients received angiotensin-converting enzyme inhibitor (ACE-I) and 32% of them were also treated with beta-blockers at the discretion of the treating physician. Patients with heart failure received intravenous immunoglobulin, milrinone, dobutamine, furosemide, and/or steroids. One patient was treated with antibiotics and another one with oseltamivir.

The disease course was favorable in all patients with complete clinical recovery. At the time of the second CMR, none of them complained of any symptoms or had any ECG changes or elevated cardiac or inflammatory markers. There was a significant improvement in their left ventricular ejection fraction (LVEF), T1 and T2 times, and a number of segments showing edema (T2W) and late gadolinium enhancement (LGE), as presented in Table 2. Left ventricular end-diastolic volume did not change between the baseline and follow-up CMR. Individual changes in LVEF, T1 and T2 times, and LGE between the baseline and follow-up CMR are presented in Figure 1.

At baseline, all patients fulfilled both the old and new Lake Louise criteria. All of them presented with elevated T1 and T2 times on parametric imaging (>52 ms and >1086 ms, respectively) as well as an elevated T2W signal intensity ratio of the left ventricular myocardium to skeletal muscle ≥2.0 and the presence of late gadolinium enhancement. According to the old Lake Louise criteria, at follow-up, a complete resolution of changes was observed in only four patients (13%). Seven patients (23%) still presented active changes and the remaining twenty (64%) showed a persistent scar. When new Lake Louise criteria were applied, incorporating parametric imaging, an additional two patients (6%) were found to have persistent active changes as defined by their increased T2 times instead of a persistent scar. The patients’ status at follow-up and changes in disease activity status according to the old and new Lake Louise criteria are presented in Figure 2.

Finally, we have looked for predictors of persistent active changes in follow-up CMR. For this reason, we have combined patients with complete recovery or the presence of a persistent scar into one group (22 patients, 71%) and compared them to those with the presence of active inflammation according to the new Lake Louise criteria (9 patients, 29%). None of the parameters except the patient’s age was predictive of persistent active changes (Table 3). Interestingly, all three patients with heart failure at baseline were free from inflammation in their CMR at follow-up, but this was not significant. Patients with persistent changes were older than those who showed a cessation of inflammatory changes, as demonstrated in Figure 3. None of the patients below 14 years of age presented with persistent active changes at follow-up and all the patients who showed persistent inflammation were between 14 and 17 years old, with the majority of them being 17 years of age (56%).

## 4. Discussion

We have confirmed, with the use of both old and new Lake Louise criteria, that despite marked improvement in left ventricular function, a decrease in T1 and T2 times and reduction in the number of segments with signs of edema or LGE, a substantial proportion of children with acute myocarditis meet the CMR criteria of active disease after a median of 6 months from their diagnosis. In fact, the application of the new Lake Louise criteria further increased that number due to the use of T2 mapping, which has been shown to be more sensitive in the detection of edema than old T2W sequences [8,14]. In our study, the use of parametric imaging was able to detect an additional two cases that displayed an active inflammatory process. Like previous studies, we have also demonstrated that the largest group of patients at follow-up consists of those showing a persistent scar in their myocardium, while only a minority of patients present complete recovery [7,8,9,10,11,12]. We and others have previously demonstrated similar observations, but these older studies did not systematically use new Lake Louise criteria and were mainly based on old criteria [9,10,11,12].

The clinical significance of these findings is currently unknown and remains a subject of investigation [1]. Like in other studies, all patients until their second CMR were free from symptoms and had normalized cardiac and inflammatory markers, a normal ECG and normal echocardiograms. They also did not present any severe arrythmia. We can only speculate about potential reasons and therefore the significance of the observed findings. First, it may be a purely imaging-based phenomenon related to the oversensitivity of the CMR criteria used to diagnose acute myocarditis, especially in their separation from the clinical picture. Studies comparing old and new Lake Louise criteria to endomyocardial biopsy have found a rate of over 10% false positive findings where the CMR was significant for acute myocarditis but an endomyocardial biopsy turned out negative [15,16]. Prolonged active inflammatory changes in CMR may also reflect an autoimmune reaction secondary to initial viral infection which persists after the infection’s cessation. It has been previously shown by us and other groups that changes typical for myocarditis may accompany autoimmune diseases [17,18]. Immune-mediated inflammation has also been proposed as a mechanism of post-COVID-19 mRNA vaccine myocarditis, which often affects older teenagers [19,20,21]. In our study, all cases of prolonged active changes were visible in older teenagers and none in children below 14 years old, which may further support this hypothesis. In fact, older age was the only predictor of persistent active changes in our study. Previous research pointed to other factors as predictors of recovery, such as a fever on admission and the subepicardial pattern of LGE [9,11]. We were not able to confirm this in our study.

Another question that is raised is how to react to persistent active findings in CMR. In our group, patients were kept on medications (ACE-I, beta-blockers) and scheduled for another CMR in 6–12 months’ time, which, as demonstrated in our previous study, usually shows a cessation of active changes, except in patients with chronic autoimmunologic diseases [7]. In the meantime, patients were also asked to refrain from more than recreational, light-to-moderate physical activity, as full myocarditis resolution including CMR is one of the criteria which should be taken into consideration before a full return to play [22,23]. Based on our results, it seems reasonable to postpone a follow-up CMR in older children by a few more months unless there is a need for an earlier check-up, such as in junior athletes. Early follow-up in that group may come at the expense of the need for more than one CMR follow-up study. Persistent areas of LGE, frequently observed at follow-up, as also demonstrated in our study, are usually not a contraindication to a return to play if there are no accompanying left ventricular dysfunctions or major arrhythmias at rest and during exercise [22,23]. Each case should be considered individually, as there are no prospective long-term studies in this group. Previous studies confirmed that even frequent premature ventricular contractions (PVCs) in children may be addressed as a benign condition and should not preclude sport participation if they are not associated with cardiac malformations, heart dysfunction or cardiomyopathy [24,25]. However, in case of a history of myocarditis, newly discovered PVCs should always prompt an additional evaluation as they may be related to persistent scars.

We did not encounter any clinical events in our group during follow-up. Studies analyzing clinical predictors point out risk factors such as the presence of an LVEF < 30% on admission, a younger age, a prolonged course of the disease and N-terminal pro-brain natriuretic peptide levels or CMR late enhancement [3,4]. This shows that markers of clinical prognosis and CMR prognosis may be very different and determining the relation between these two entities requires prospective, multi-center studies on large sample sizes.

Our study has some limitations. We did not use an endomyocardial biopsy to confirm the presence of myocarditis. However, like in many other institutions, in an uncomplicated course, the diagnosis of myocarditis is usually based on the clinical picture, ECG, biomarkers and CMR [1]. Secondly, our study had a retrospective nature and was based on a relatively small sample size, which might have affected the results. However, we have analyzed all consecutive patients admitted to the hospital; all of them had undergone two CMR studies with the same protocol using parametric imaging, which was not systematically carried out in previous studies.

## 5. Conclusions

We have shown that pediatric myocarditis can lead to persistent active changes in CMR beyond the typical timeframe of the control 6-month follow-up in over a fifth of patients. The application of more sensitive new Lake Louise criteria further increases that number in comparison to the old criteria. The only predictor of persistent inflammatory changes in CMR in our study was the older age of children. However, this initial finding and its clinical significance require further studies.

## Figures and Tables

**Figure 1 jcm-13-06498-f001:**
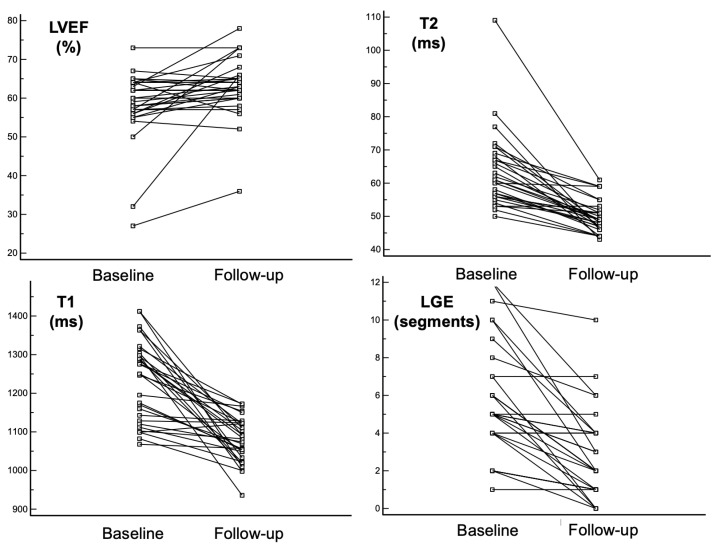
Individual changes in CMR parameters from baseline to follow-up. CMR—cardiac magnetic resonance; LGE—late gadolinium enhancement; LVEF-left ventricular ejection fraction.

**Figure 2 jcm-13-06498-f002:**
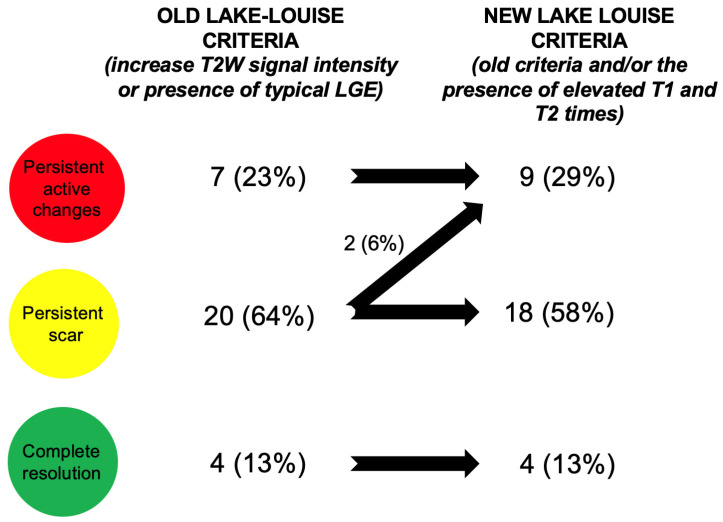
Patient status and “migration” trends on follow-up CMR according to old and new Lake Louise criteria. LGE—late gadolinium enhancement; T2W—T2-weighted images.

**Figure 3 jcm-13-06498-f003:**
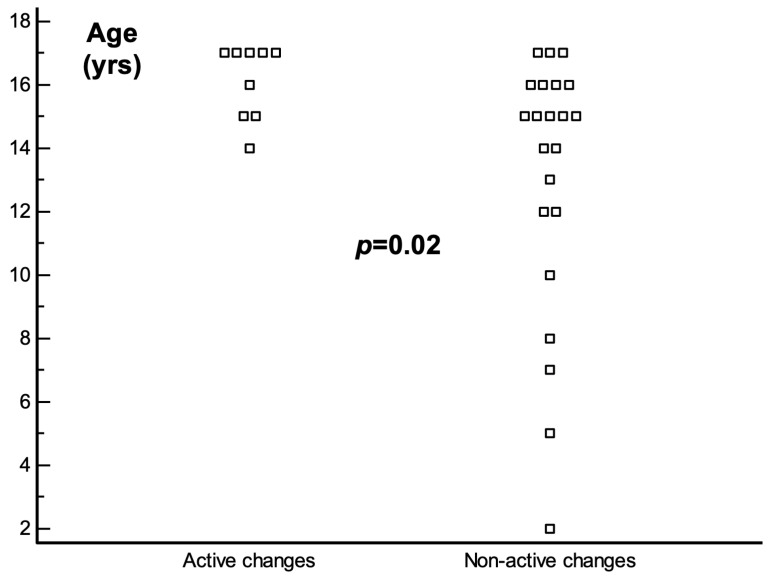
Older age as a predictor of persistent active changes at follow-up CMR. CMR—cardiac magnetic resonance; LGE—late gadolinium enhancement; LVEF-left ventricular ejection fraction.

**Table 1 jcm-13-06498-t001:** Baseline characteristics of patients.

Parameter	
Age, yrs	15 (13.3–16.8)
Male sex	21 (68)
BSA	1.7 (1.5–2.0)
Symptoms	
Fever	11 (35)
Chest pain	22 (71)
Heart failure	3 (10)
ECG changes	14 (45)
Troponin I, ng/mL	707 (12–3264)
Medications	
ACE-I	31 (100)
Beta-blocker	10 (32)
IVIG	3 (10)
Dobutamine	2 (6)
Milrinone	2 (6)
Furosemide	2 (6)
Steroids	2 (6)
Antibiotic	1 (3)
Oseltamivir	1 (3)
LVEDVI, mL/m^2^	83 ± 19
LVEF, %	59 (56–63.8)
Time between first and second CMR	6 (6–7.8)

ACE-I—angiotensin-converting enzyme inhibitor; BSA—body surface area; CMR—cardiac magnetic resonance; ECG—electrocardiogram (ST elevation, T-wave inversion); IVIG—intravenous immunoglobulin; LVEDVI—left ventricular end-diastolic volume index; LVEF—left ventricular ejection fraction.

**Table 2 jcm-13-06498-t002:** Differences between initial and follow-up CMR.

Parameter	Baseline CMR Study	Follow-Up CMR Study	*p*
LVEDVI, mL/m^2^	83 ± 19	83 ± 17	0.98
LVEF, %	59.5 (56–64)	62.5 (60–66)	0.0025
LGE, no. of segments	5 (4–7)	2 (1–4)	<0.0001
T2W, no. of segments	6 (4–15.8)	0 (0–0.75)	<0.0001
T1 time, ms	1234 ± 105	1077 ± 59	<0.0001
T2 time, ms	61 (56–67.8)	49 (47–51.8)	<0.0001

CMR—cardiac magnetic resonance; LGE—late gadolinium enhancement; LVEDVI—left ventricular end-diastolic volume index; LVEF—left ventricular ejection fraction; T2W—T2-weighted images.

**Table 3 jcm-13-06498-t003:** Patient characteristics according to the presence or absence of ongoing active changes at follow-up CMR.

Parameter	Active Changes N = 9	Non-Active Changes N = 22	*p*
Age, yrs	17 (15–17)	15 (12–16)	0.02
Male sex	5 (56)	16 (73)	0.42
Time between first and second CMR	6 (6–8.8)	6.5 (6–7)	0.97
Fever	4 (44)	7 (32)	0.68
Chest pain	7 (78)	15 (68)	0.69
Heart failure	0 (0)	3 (14)	0.54
ECG changes	5 (56)	9 (41)	0.69
Troponin I, ng/mL	3152 (0.1–4380)	659 (10–2625)	0.46
LVEDVI, mL/m^2^	79 (70–83)	84 (73–91)	0.21
LVEF, %	62 (57–64)	63 (60–69)	0.27
LGE, no. of segments	5 (4.8–6.5)	5 (4–7)	0.74
LGE pattern than subepicardial	5 (56)	15 (68)	0.68
LGE localization			
infero-lateral	8 (89)	14 (64)	0.22
other	5 (56)	14 (64)	0.70
T2W, no. of segments	8 (4.8–17)	5.5 (4–12)	0.38
T1, ms	1250 (1109–1332)	1262 (1143–1298)	0.93
T2, ms	61 (55–70)	60 (56–67)	0.76

CMR—cardiac magnetic resonance; ECG—electrocardiogram; LGE—late gadolinium enhancement; LVEDVI—left ventricular end-diastolic volume index; LVEF—left ventricular ejection fraction; T2W—T2-weighted images.

## Data Availability

The data are available upon request.

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
