# Peer review of "Older Age as a Predictor of Ongoing Active Changes in Follow-Up Cardiac Magnetic Resonance in Children with Acute Myocarditis"

_jcm, 2024, doi:10.3390/jcm13216498_

Round 1

Reviewer 1 Report

Comments and Suggestions for Authors

This is a study of great clinical significance, indicating that older age serves as a predictor of persistent inflammatory changes on 26 CMR. However, the conclusion is not fully reliable due to the small sample size of only 31 children in the study. It is recommended to include more samples and perform correlation analysis to draw more reliable conclusions

Comments on the Quality of English Language

It is recommended to make appropriate revisions to the English

Author Response

Thank you very much for your comments. I have reviewed the manuscript and performed the changes (marked in review mode in text). My answers to the commments and the consequent changes are described in detail below.

Reviewer 1 comments:

This is a study of great clinical significance, indicating that older age serves as a predictor of persistent inflammatory changes on 26 CMR. However, the conclusion is not fully reliable due to the small sample size of only 31 children in the study. It is recommended to include more samples and perform correlation analysis to draw more reliable conclusions

Thank you for the kind words regarding the importance of our findings. Our study was based on all consecutive CMR studies with proven myocarditis from a single center. It is based on 31, not 26 CMR studies. Inclusion of more samples would delay the publication of those important results, which, as we feel, is methodologically justified for the following reasons:

  1. Older age as a risk factor was a stand-alone finding with not a marginal significance (p=0.02) and the difference is clearly visible in Figure 3.
  2. All our patients underwent follow-up study, with no missing studies, which was a case for many previous retrospective studies. We have used a more robust new Lake-Louise criteria and not the old ones.
  3. We have presented a plausible hypothesis for the findings in the discussion section.
  4. Correlation would not necessarily bring more insight as this phenomenon may be more dichotomous than linear with children above certain age more likely to presen persisted active changes (for example 13-14 years and above?). Following your comment, we have run correlations between age and markers of active changes on follow-up CMR (T2 ratio, T2 and T1 tines and LGE) and found only some weak correlations for T2 time (rho=0.35, p=0.06) and the number of segments with T2 ratio >2 (rho=0.43, p=0.02), which we decided not to include in the manuscript.
  5. Our study was based on similar number of patients as most other single center reports from the last period searching for risk factors of persistent active myocarditis on follow-up CMR (for example Ait-Ali L et al – 50 patients, 40 vs. 10, Isaak A et al. 56 patients, 43 vs 13, but only 27 patients underwent follow-up CMR). Larger groups were typical only for multicenter studies.
  6. Finally, our previous study presenting large percentage of persistent active changes on CMR in children with myocarditis was based on almost two times smaller sample (18 patients) and despite that it was considered as sound with 26 citations and inclusion in AHA guidelines (doi: 10.1002/jmri.27036., doi: 10.1161/CIR.0000000000001001.).

However, following your comment we have decided to soften the conclusions by adding the following sentence: “The only predictor of persistent inflammatory changes on CMR in our study was older age of children. However, this initial finding and its clinical significance require further studies.”

Reviewer 2 Report

Comments and Suggestions for Authors

However, the paper is very interesting and the discussion should be enriched. What could be the scenario of these patients in future life. For example, sporting activity??? read  https://doi.org/10.2459/JCM.0000000000001186

Comments on the Quality of English Language

Good

Author Response

Thank you very much for your comments. I have reviewed the manuscript and performed the changes (marked in review mode in text). My answers to the commments and the consequent changes are described in detail below.

However, the paper is very interesting and the discussion should be enriched. What could be the scenario of these patients in future life. For example, sporting activity??? read  https://doi.org/10.2459/JCM.0000000000001186

            Thank you for the comment and an insight that our study, as we also believe is very interesting. We have added the suggested citation to the discussion section along already existing citation of ESC guidelines in sports cardiology also from 2020. We have already described the scenario of patients with persistent changes in context of their return to play in the discussion section: “Another question is how to react to persistent active findings on CMR. In our group patients were kept on medications (ACE-I, beta-blockers) and scheduled for another CMR at 6–12-month time, which as demonstrated in our previous study, usually shows cessation of active changes except in patients with chronic autoimmunologic diseases [7]. In a meantime, patients were also asked to refrain from other than recreational, light to moderate physical activity as full myocarditis resolution including CMR is one of the criteria which should be taken into consideration before a full return to play [22, new citation]. Based on our results seems reasonable to postpone follow-up CMR in older children by a few more months unless there is a need for earlier check-up as in junior athletes. However, early follow-up in that group may come at the expense of the need for more than one CMR follow-up study.”  Currently we have added an additional sentence: ”Persistent areas of LGE, frequently observed at follow-up as demonstrated also in our study, are usually not a contraindication to return to play if there are no accompanying left ventricular dysfunction or major arrhythmias at rest and during exercise [22, new citation]. However, each case should be considered individually as there are no prospective long-term studies in that group.”  

Reviewer 3 Report

Comments and Suggestions for Authors

This study is extremely interesting, focusing on pediatric myocarditis that may lead to persistent active changes in CMR beyond 6 months follow-up in over a fifth of patients. According to results, the application of new Lake-Louise criteria further increases that number compared to the old criteria: the only predictor of persistent inflammatory changes on CMR is older age.I have only few minor comments in order to improve the manuscript.

In particular, authors should more discuss about the importance of arrhythmic burdern  in particular, premature ventricular complexes (PVCs) are frequently documented in children. To date, few studies report long-term follow-up in pediatric cohorts presenting with frequent PVCs. In this scenario a careful follow-up is mandatory, and more in case of onset of symptoms and/or ECG/echocardiographic changes  (DOI: 10.1007/s00246-019-02233-w). Please cite suggested reference  and amplify the discussion

At the same time authors should more discuss about impact of Covid-19.  Covid-19 has confused the whole world, and myocarditis may be a dangerous complication of Covid-19 also in pediatric population. Because Covid-19 and its side effects as myocarditis are very new to the world there is a need for further research to overcome them. (DOI: 10.1002/hsr2.488). Please cite suggested reference and amplify the discussion

Please discuss also relationship with vaccination Please cite doi: 10.3389/fcvm.2022.951314

Author Response

Thank you very much for your comments. I have reviewed the manuscript and performed the changes (marked in review mode in text). My answers to the commments and the consequent changes are described in detail below.

This study is extremely interesting, focusing on pediatric myocarditis that may lead to persistent active changes in CMR beyond 6 months follow-up in over a fifth of patients. According to results, the application of new Lake-Louise criteria further increases that number compared to the old criteria: the only predictor of persistent inflammatory changes on CMR is older age. I have only few minor comments in order to improve the manuscript.

            Thank you for the positive feedback on our results and all the comments addressed individually below.

In particular, authors should more discuss about the importance of arrhythmic burdern  in particular, premature ventricular complexes (PVCs) are frequently documented in children. To date, few studies report long-term follow-up in pediatric cohorts presenting with frequent PVCs. In this scenario a careful follow-up is mandatory, and more in case of onset of symptoms and/or ECG/echocardiographic changes  (DOI: 10.1007/s00246-019-02233-w). Please cite suggested reference and amplify the discussion

             Thank you for the comment. We have dedicated a different study to this subject before (J Clin Med. 2021 Mar 24;10(7):1335. doi: 10.3390/jcm10071335.). Following your comment however, we have decided to cite the suggested paper as well as our study and include additional sentences in the discussion section on the return to play after myocarditis: “Previous studies confirmed that even frequent premature ventricular contractions (PVCs) in children may be addressed as a benign condition and should not preclude sport participation if are not associated with cardiac malformations, heart dysfunction, or cardiomyopathy [new citations]. However, in case of a history of myocarditis newly discovered PVCs should always prompt additional evaluation as they may be related to persistent scars.” 

At the same time authors should more discuss about impact of Covid-19.  Covid-19 has confused the whole world, and myocarditis may be a dangerous complication of Covid-19 also in pediatric population. Because Covid-19 and its side effects as myocarditis are very new to the world there is a need for further research to overcome them. (DOI: 10.1002/hsr2.488). Please cite suggested reference and amplify the discussion

            Thank you for that comment. We have specifically excluded all cases of myocarditis related to the vaccination or Covid-19. As one of the first ones, we have addressed the issue of Covid-19 related complications in children in CMR in the different paper (J Magn Reson Imaging. 2022;55(3):883-891. doi: 10.1002/jmri.27870). Therefore, to clarify the message of the current study, we have decided not to include this study in the discussion. 

Please discuss also relationship with vaccination Please cite doi: 10.3389/fcvm.2022.951314

             Thank you for the comment. Our study did not include patients with post-Covid-19 vaccination myocarditis. All these cases have been excluded as mentioned in the Materials and methods section. However, we have mentioned this form of myocarditis in the Discussion section as it is believed to be auto-immunologically mediated and occurs in more often in older children. The following sentence were present in the Discussion section which were enriched with the new citation suggested above: “Immune-mediated inflammation has been also proposed as a mechanism of post-Covid-19 mRNA vaccine myocarditis, which often affects older teenagers [19, 20, new citation].”

Round 2

Reviewer 3 Report

Comments and Suggestions for Authors

Manuscript definitely improved. Congratulations